# Perioperative Blood Transfusion Is Dose-Dependently Associated with Cancer Recurrence and Mortality after Head and Neck Cancer Surgery

**DOI:** 10.3390/cancers15010099

**Published:** 2022-12-23

**Authors:** Hui-Zen Hee, Kuang-Yi Chang, Chii-Yuan Huang, Wen-Kuei Chang, Mei-Yung Tsou, Shih-Pin Lin

**Affiliations:** 1Department of Anesthesiology, Taipei Veterans General Hospital, Taipei 112201, Taiwan; 2School of Medicine, National Yang Ming Chiao Tung University, Taipei 112201, Taiwan; 3Department of Otorhinolaryngology-Head and Neck Surgery, Taipei Veterans General Hospital, Taipei 112201, Taiwan

**Keywords:** head and neck cancer, blood transfusion, cancer recurrence, second primary cancers, survival analysis

## Abstract

**Simple Summary:**

In head and neck cancer surgery, blood transfusion is required occasionally due to patients’ underlying conditions and perioperative blood loss during surgical resection. However, transfusion is associated with immunosuppression, also known as the term “transfusion-related immune modulation (TRIM)”, which could lead to worse cancer prognoses. The purpose of the study is to assess the association between perioperative blood transfusion and head and neck cancer recurrence and mortality. Our findings showed that blood transfusion was significantly associated with both cancer recurrence and mortality after head and neck cancer surgery.

**Abstract:**

Background: The association between perioperative blood transfusion and cancer prognosis in patients with head and neck cancer (HNC) receiving surgery remains controversial. Methods: We designed a retrospective observational study of patients with HNC undergoing tumor resection surgery from 2014 to 2017 and followed them up until June 2020. An inverse probability of treatment weighting (IPTW) was applied to balance baseline patient characteristics in the exposed and unexposed groups. COX regression was used for the evaluation of tumor recurrence and overall survival. Results: A total of 683 patients were included; 192 of them (28.1%) received perioperative packed RBC transfusion. Perioperative blood transfusion was significantly associated with HNC recurrence (IPTW adjusted HR: 1.37, 95% CI: 1.1–1.7, *p* = 0.006) and all-cause mortality (IPTW adjusted HR: 1.37, 95% CI: 1.07–1.74, *p* = 0.011). Otherwise, there was an increased association with cancer recurrence in a dose-dependent manner. Conclusion: Perioperative transfusion was associated with cancer recurrence and mortality after HNC tumor surgery.

## 1. Introduction

Head and neck carcinoma is a type of cancer that affects the tissues of the head and neck, including the mouth, nose, and throat. It is the seventh most common of all cancers and can be caused by a variety of factors, including tobacco and alcohol use, exposure to certain chemicals, and infection with human papillomavirus (HPV) [1,2]. There are more than 890,000 new cases and 450,000 deaths each year. The incidence of HNC was high in Taiwan because of the culture of betel nut chewing, tobacco use, and alcohol consumption. The growing incidence rate in Taiwan has brought attention to the refinement of the treatment, including strategies for minimizing future recurrences [3,4]. In locally advanced (stage III and IV) HNCs, the locoregional recurrence rate is about 60% within 2 years following resection, chemotherapy, radiotherapy, or any of the combinations above. Meanwhile, the distant metastases rate is about 20% to 30%, and the risk of having another primary tumor is about 10% to 20% [5,6].

Factors associated with tumor recurrence are mainly related to tumor characteristics [7]. Given the underlying anemic condition due to tumor growth or patient malnutrition and inevitable perioperative blood loss during surgical resection, a blood transfusion might occasionally be required. Some studies suggested blood transfusion as an independent risk factor and predictor for tumor recurrence and survival [8,9,10,11]. Other predictors, such as surgical margin, cancer stage, preoperative hemoglobin levels, and age, were also identified. However, controversies do exist and retrospective cohort studies have a risk of selection bias that will affect the main treatment effect because of the nature of the study design [12]. Herein, in this study, the inverse probability of treatment weighting (IPTW) method was applied to minimize this bias by balancing the known confounders that could affect the treatment effect without reducing the sample size. The goal of this study is to determine whether blood transfusion increases the risk of HNC recurrence and mortality in a dose-dependent manner.

## 2. Materials and Methods

### 2.1. Inclusion and Exclusion Criteria

This study was approved by the Institutional Review Board (IRB-TPEVGH no. 2017-12-025BC). We retrospectively collected data from patients who received surgery for head and neck surgery at Taipei Veterans General Hospital, a tertiary medical center, from January 2014 to December 2017. Patients with previous head and neck cancer surgery or missing key study data (such as patients’ characteristics, surgical, anesthetic and pathologic records) were excluded from the study.

### 2.2. Data Collection

All data were collected from electronic medical records. The collected variables include the demographic characteristics, American Society of Anesthesiologists (ASA) classification, history of smoking habits, betel nut chewing and alcohol consumption, preoperative hemoglobin value, anesthesia time length, total blood loss during surgery, perioperative packed red blood cell (pRBC) transfusion amount, histological findings of surgical margin, tumor metastasis, staging classification, histological differentiation and whether postoperative adjuvant radiotherapy or chemotherapy was received. Perioperative blood transfusion was defined as an allogeneic transfusion of packed red blood cells during surgery or within 7 days after the operation. We converted TMN staging into stages I to IV according to the American Joint Committee on Cancer criteria (AJCC-7). The date of mortality was retrieved from medical records. Cancer recurrence and diagnosis of a second primary cancer were by other independent radiologists and surgeons based on imaging studies (e.g., computerized tomography, magnetic resonance imaging, bone scan, etc.) or pathological proof from a tissue biopsy. Our primary outcome was recurrence-free survival (RFS), which was defined as the time from the date of surgery to the date of tumor recurrence. The secondary outcome was overall survival (OS), defined as the time from the date of surgery to the date of death, and the occurrence of a second primary cancer, defined as the time from the date of surgery to the date of diagnosis of a second primary cancer. Those patients who did not have tumor recurrence or death were treated as a censored variable in the survival analysis.

### 2.3. Statistical Analysis

All patients were divided into two groups based on whether they received perioperative pRBC transfusion or not. Continuous variables are presented as mean with standard deviation and categorical variables are presented as count with percentage. If continuous variables did not fit normal distribution (such as anesthetic time or blood loss amount), logarithmic transformation was used to reduce the skewness of the distribution. Standardized differences are used to evaluate the balance between the two groups, which calculates the mean difference between the two groups divided by an estimate of the standard deviation. Inverse probability treatment weighting (IPTW) is a statistical method used to adjust for bias in observational studies. It is commonly used in medical research to account for differences between the study groups that may affect the validity of the study results. In an IPTW analysis, each study participant is assigned a weight based on the probability that they were included in the study. The individual’s probability of weighting is the probability of a list of documented confounding factors affecting blood transfusion (Appendix A) and is calculated as an individual’s propensity score. This weight is then inversely used to adjust the analysis so that the study groups are more similar to each other, which helps to reduce bias and improve the validity of the study results. This means that samples that are less likely to be selected are given more weight, and samples that are more likely to be selected are given less weight. The goal of inverse probability weighting is to account for the sampling design and to reduce bias in the estimated population average [13]. These inverse probabilities are then weighted in the regression analysis so that the study groups are more similar to each other, which helps to reduce bias and improve the validity of the study. The Kaplan–Meier method is used for estimating the proportion of patients who are alive for a certain period of time after being diagnosed with a disease (survival time). The Kaplan–Meier method is a non-parametric method for estimating the survival function from tumor recurrence and the occurrence of a second primary tumor. The Cox regression model was applied to analyze the covariate effects on the risk of tumor recurrence, overall survival, and risk of a second primary cancer with censored observation. A restricted cubic spline (RCS) is used to model non-linear relationships between transfusion dosage and tumor outcome. It is a type of regression analysis that uses a set of basis functions to fit a smooth curve to the data. The “restricted” part of the name refers to the fact that the function is constrained to pass through a set of pre-specified knots, which helps to prevent overfitting and improve the interpretability of the model. The cubic spline part of the name refers to the fact that the function is defined by a set of cubic polynomial segments that are joined together at the knots. RCS functions are commonly used in medical research to model complex relationships between predictor and outcome variables. For multivariate analysis, we use stepwise model selection for those significant predictors of recurrence-free or overall survival. The significance level for all hypotheses was 0.05 for a two-tailed test. All the statistical analyses were performed using SAS software, version 9.4 (SAS Institute Inc., Cary, NC, USA).

## 3. Results

A total of 683 patients were included in the study; 192 (28.1%) of them received a perioperative blood transfusion in the perioperative period. The detailed data are presented in the Appendix A. In the study population, patients who received perioperative blood transfusion had more advanced cancer, greater blood loss, longer anesthesia time and higher rates of lymph node involvement, and a higher requirement for adjuvant postoperative radiotherapy or chemotherapy. We noticed the imbalanced allocations between the two groups, and after IPTW, these imbalances of these major prognostic factors between the two groups were greatly improved (Table 1).

### 3.1. Perioperative Transfusion and Recurrence Risk

Perioperative pRBC transfusion was shown to be associated with recurrence risk (crude hazard ratio (HR) = 1.69, *p* < 0.001, Figure 1A) in the univariate analysis. After IPTW weighting, the risk of the perioperative transfusion and postoperative head and neck cancer recurrence is still significant (adjusted HR = 1.37, 95% confidence interval [CI] 1.1 to 1.7; *p* = 0.006, Figure 2A) in the weighted Cox regression analysis. Further analysis of the dosage of blood transfusion and tumor recurrence demonstrated a significant non-linear dose–response association between pRBC transfusion and cancer recurrence after surgery (Table 2). The risk of blood transfusion showed a concave that increased and peaked at around 6 units and decreased gradually thereafter (Figure 3A). There are four independent predictors of cancer recurrence, including positive surgical margin (HR = 1.63), advanced cancer stage (HR = 1.59), lymph node involvement (HR = 1.96), and adjunct radiotherapy (HR = 0.61) (Table 3) after multivariate analysis. The association of perioperative transfusion and cancer recurrence after surgery for head and neck cancer was of borderline significance (adjusted HR = 1.4, 95% CI: 0.99~1.97, *p* = 0.059) after the adjustment for these significant predictors.

### 3.2. The risk of Perioperative Transfusion and Mortality

Perioperative pRBC transfusion showed an increased risk of overall survival after head and neck cancer surgery (crude HR = 2.37, *p* < 0.001, Figure 1B). There is a significant association between blood transfusion and inferior overall survival (adjusted HR: 1.37, 95% CI: 1.07–1.74, *p* = 0.011, Figure 2B) after IPTW. There was a significant linear dose-dependent pRBC transfusion risk to overall survival (Table 2) and the mortality risk. The risk gradually increased with the amount of pRBC transfusion (Figure 3B). Four independent factors of overall survival were identified, including age (HR = 1.02), BMI (HR = 0.94), advanced cancer stage (HR = 1.77), and lymph node involvement (HR = 3.44) (Table 4). The effect of perioperative transfusion on overall survival after head and neck cancer surgery was of borderline significance in the final model (adjusted HR = 1.39, 95% CI: 0.98 to 1.97, *p* = 0.067).

### 3.3. Perioperative Transfusion and Occurrence of a Second Primary Cancer

The distribution of the occurrence of second primary cancers is presented in Appendix A. There was no significant association between the occurrence of a second primary cancer and perioperative pRBC transfusion (crude HR = 1.355, *p* = 0.164, Figure 1C) in the univariate analysis. After IPTW, the relationship between transfusion and the occurrence of second primary cancer remained non-significant (adjusted HR = 1.29, 95% CI: 0.97 to 1.71, *p* = 0.085, Figure 2C). Further regression analysis identified only one risk factor, alcohol (HR = 2.69, 95% CI: 1.665 to 4.39, *p* = 0.007).

## 4. Discussion

The result of the study showed that perioperative pRBC transfusion is associated with an increase in both the tumor recurrence and mortality of head and neck tumors. This relationship occurs in a dose-dependent manner, with non-linear relationship for recurrence and linear for mortality. The implementation of the IPTW method, which is relatively novel in the discussed topic, created a weighted sample where the distribution of the covariates is equal between those who were transfused and those who were not. Because of this, not only did we reduce the imbalances in patient characteristics but also preserved the sample size and statistical power; this provided a more precise and accurate estimation of the treatment effect [14]. Additionally, the dose-dependent effect of perioperative packed red blood cell (pRBC) transfusion may further strengthen the association between tumor recurrence or mortality following resection of head and neck cancer [15].

Previous investigations have been carried out in search of the connection between blood transfusion and head and neck cancer recurrence; the outcomes remain controversial. In a multivariate analysis conducted by Perisanidis et al., a detrimental association between perioperative blood transfusions and the postoperative complication rate in 142 patients diagnosed with oral and oropharyngeal squamous cell carcinoma (complication rate of 84% vs. 39%, *p* < 0.001) was demonstrated, but the recurrence rate was not significant (recurrence-free survival probability at 5 years of 49% vs. 62%, log rank *p* = 0.44) [16,17]. In another cohort study including 223 patients by Fenner et al., the transfusion of >4 units of blood did not have a significant impact on the overall survival in patients receiving primary surgery for oropharyngeal carcinoma (RR = 1.53, 95%CI 0.84–2.81, *p* = 0.17) [18]. Reviewing the literatures, with their relatively limited sample size and interference of confounders being the major limitation, the application of the IPTW method could provide a valuable method for eliminating those factors.

Reviewing recent studies consistent with our findings, Chau et al. reported a correlation between higher recurrence rates and decreased survival with transfusion of leukodepleted blood perioperatively in a study on 520 head and neck cancer patients receiving surgery [8]. Another retrospective study by Szakmany et al. included 559 patients undergoing primary surgery for oral and oropharyngeal squamous cell carcinoma and revealed that pRBC transfusion of more than 3 units is associated with tumor recurrence (29%, 19%, and 19% for the rate of recurrence for patients transfused with 3 or more units, 1–2 units, and no transfusion, respectively, *p* = 0.06 χ2 test) [9]. The relatively large sample sizes of the aforementioned research represent one of the merits of their studies, and to balance the confounding effects, multivariate analysis was used. Our current study further supports the previous investigations, where we applied another approach by using a propensity score probability as an inverse weighting to minimize the imbalance distribution between two groups. A significant improvement was shown after IPTW by reducing the absolute standardized differences. This provided comparable groups to assess the effect of blood transfusion. By simulating a randomized control trial scenario, IPTW provided a more intuitive and useful way to evaluate the treatment effect by using known confounding factors, especially in retrospective studies [14,19].

The most commonly proposed mechanism contributing to the influences of allogenic blood transfusion (ABT) on cancer outcome is the phenomenon known as transfusion-related immunomodulation (TRIM), which refers to the immune alterations associated with allogenic blood transfusion [20]. In a detailed review article by Tzounakas et al., the authors described the underlying mechanisms of TRIM, including RBC storage lesions, the RBCs themselves, residual white blood cells, immunosuppressive cytokines, and biologic mediators [21]. Another nontypical immunomodulator, the extracellular vesicles (EVs) are RBC storage lesions that increase in amount as the RBCs age. EVs can be secreted by all cell types, and their biological functions include eliminating cellular waste, facilitating intercellular communication, and, of particular interest, modulating the immune response [22]. The contents within EVs secreted by RBCs including RNAs, immunoglobulins, complement proteins, and exposed phosphatidylserine are believed to actively play a role in activating TRIM in cancer patients [23]. The abovementioned components in the blood products could elicit immune responses that include both immunosuppressive and inflammatory effects. The underlying pathophysiology is complicated, involving the suppression of monocytes, cytotoxic T cells, and NK cell activity, as well as the inhibition of interleukin-2(IL-2) production [24,25], whose roles are crucial in the process of cancer cell recognition and eradication. In adjunction to the suppressant effects, ABT causes the amplification of regulatory T cells and immunosuppressive prostaglandins activities, which in turn suppresses the Th1 response accountable for the subsequent release of cytokines that activate death receptors on the tumor cell surface [26].

These reactions contributed to the postulated theory for unopposed tumor cell proliferation and dissemination, with sequential metastatic spread, which is compatible with our study result. The effect of TRIM on tumor outcomes has also been implemented in several cancerous diseases, including hepatocellular carcinoma [27], colorectal carcinoma [28], cervical carcinoma [29], esophageal carcinoma [30], and non-small cell lung cancers [31]. The IPTW method has been applied in some of the studies [32,33,34], yielding significant and convincing results. However, to date, there were none among them using IPTW in researching the effects of blood transfusion on the recurrence of head and neck cancers.

Several variables with prognostic importance have been proved by previous authors, such as age, tumor stage, lymph node involvement, completeness of resection, etc. Our findings further supported those conclusions and shed light on another factor, the body mass index (BMI). After IPTW in our study, it is shown that a lower BMI is independently associated with worse overall survival (Table 4), and intuitively, we suggest that a balanced diet plays a critical role in the protective effect against cancer. It has been reported that several dietary factors including macro- and micronutrients such as vitamin C, vitamin E, and carotene may be beneficial in the outcome of cancers when included in a balanced diet [35,36]. Given the circumstances of HNC patients in their difficulties with food intake including dysphagia as well as involuntary weight loss because of cancer, by the time of diagnosis, 60% of the population has been reported to have malnutrition [37], with cachexia in one-third of the population [38]. The negative impact of malnutrition on cancerous disease has been reported in several studies [35,39], and besides a healthier immune system, certain antioxidative properties of the nutrients contribute to anticarcinogenic activity. However, the BMI recorded in our study was merely the BMI at the time of surgery, and the weight changes along the cancer treatment course were not tabulated. Nevertheless, the data could be obtained and more insightful analysis is inclined to further discussions.

One limitation of our study is its retrospective nature, which may have contributed to nonrandomized and unstandardized interventions. This type of study design does not permit the inference of causal relationships. While we carefully controlled for known confounding factors using inverse probability of treatment weighting (IPTW) analysis, residual confounding effects may still be present. Additionally, in this study, we did not examine the effect of leukodepletion on packed red blood cell (pRBC) transfusions. Most of the pRBC transfusions in our data were not leukodepleted. While the literature has examined the impact of leukocyte depletion on immunomodulation and cancer outcomes, both leukodepleted and non-leukodepleted pRBCs have been linked to poor cancer outcomes. Two randomized controlled trials found no association between leukocyte-depleted blood products and cancer recurrence or long-term survival in colorectal cancer [40] and gastrointestinal cancer patients [41]. However, another study demonstrated that the transfusion of non-leukodepleted pRBCs was associated with ovarian cancer recurrence [42]. Moreover, we did not consider the transfusion of other blood products such as fresh frozen plasma, platelet concentrates, platelet apheresis, etc., into account. Further study of leukodepleted status effects on cancer outcomes can be performed in the future. Nonetheless, we believe that given the circumstances of our study design, it could still be representative of the larger population.

## 5. Conclusions

Based on the above findings, we concluded that in head and neck cancer, recurrence and mortality were significantly associated with perioperative blood transfusion in a dose-dependent relationship. However, this study is retrospective with all the inherent shortcomings of retrospective studies, and future prospective studies are sought to be carried out for better clarity on this issue.

## Figures and Tables

**Figure 1 cancers-15-00099-f001:**
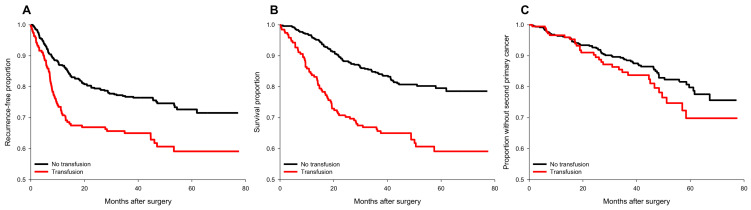
Survival plots of cancer recurrence, all-cause mortality, and occurrence of second primary cancers for transfused and non-transfused patients before IPTW. (**A**) cancer recurrence; (**B**) all-cause mortality; (**C**) occurrence of a second primary cancer. No transfusion group (black line); transfusion group (red line).

**Figure 2 cancers-15-00099-f002:**
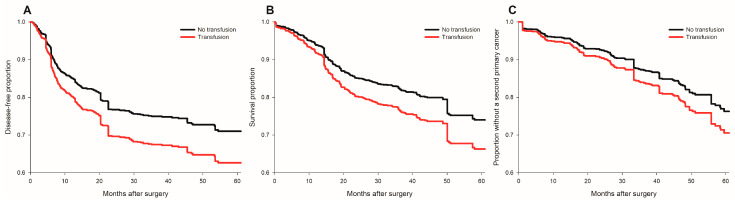
Survival plots of cancer recurrence, all-cause mortality and occurrence of second primary cancers for transfused and non-transfused patients after IPTW. (**A**) cancer recurrence; (**B**) all-cause mortality; (**C**) occurrence of a second primary cancer. No transfusion group (black line); transfusion group (red line).

**Figure 3 cancers-15-00099-f003:**
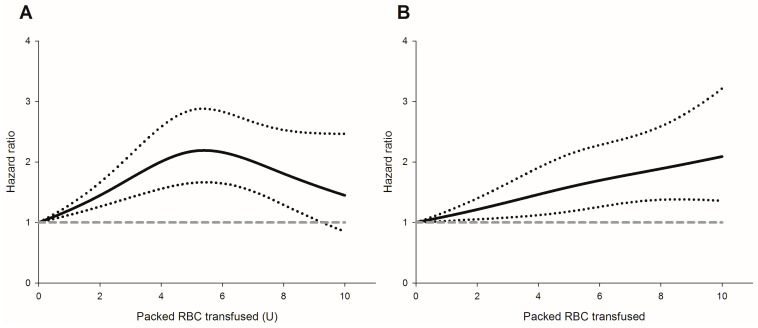
The packed red blood cell transfusion showed a dose-dependent risk for (**A**) cancer recurrence; (**B**) all-cause mortality. Hazard ratio (solid line); 95% confidence intervals (dotted lines); a reference of hazard ratio = 1 (dashed line).

**Table 1 cancers-15-00099-t001:** Baseline characteristics for patients receiving a blood transfusion or not during the perioperative period of curative resection for head and neck cancer.

	Original Data			After IPTW		
	No Transfusion (n = 491)	Transfusion (n = 192)	SDD	No Transfusion	Transfusion	SDD
Age	58 ± 13	57 ± 11	8.9	58 ± 12	57 ± 11	4.8
Sex, male	417 (84.9%)	172 (89.6%)	14.0	560 (86.8%)	434 (91.3%)	14.6
BMI, kg·m^−2^	25.0 ± 4.3	24.15 ± 3.68	21.6	24.87 ± 4.18	25.22 ± 3.93	8.5
ASA physical status > 3	112 (22.8%)	67 (34.9%)	26.9	179 (27.7%)	175 (36.9%)	19.8
Smoking	358 (72.9%)	151 (78.6%)	13.4	489 (75.8%)	396 (83.4%)	19.2
Betel nut chewing	237 (48.3%)	138 (71.9%)	49.7	354 (54.8%)	323 (68.1%)	27.5
Alcohol	293 (59.7%)	123 (64.1%)	9.0	397 (61.5%)	338 (71.1%)	20.3
Preoperative haemoglobin, g·dL^−1^	13.9 ± 1.6	13.1 ± 1.7	48.7	13.8 ± 1.7	13.5 ± 1.5	18.6
Anesthesia time, min *	8.15 ± 0.97	9.67 ± 0.71	178.4	8.49 ± 1.09	8.90 ± 1.14	36.8
Blood loss during surgery, mL *	5.60 ± 2.10	8.87 ± 1.48	179.9	6.33 ± 2.32	7.41 ± 2.15	48.5
Positive surgical margin	315 (64.2%)	113 (58.9%)	10.9	380 (58.8%)	244 (51.4%)	14.9
Primary tumour			117.6			15.1
T1 and T2	300 (61.1%)	27 (14.1%)		408 (63.3%)	265 (55.9%)	
T3 and T4	191 (38.9%)	165 (85.9%)		237 (36.7%)	210 (44.1%)	
Histologic differentiation			8.1			5.6
Well	213 (43.4%)	91 (47.4%)		305 (47.3%)	238 (50.1%)	
Moderate to severe	278 (56.6%)	101 (52.6%)		340 (52.7%)	237 (49.9%)	
Lymph node involvement	127 (25.9%)	99 (51.6%)	54.7	199 (30.8%)	183 (38.5%)	16.1
Adjunct radiotherapy	181 (36.9%)	136 (70.8%)	72.5	293 (45.4%)	228 (48.0%)	5.1
Adjunct chemotherapy	143 (29.1%)	125 (65.1%)	77.3	244 (37.9%)	207 (43.6%)	11.7

Values were mean ± SD or counts (percent). Standardized difference (SDD) is the difference in mean or proportion divided by the pooled standard error, expressed as a percentage; imbalance is defined as an absolute value greater than 20 (small effect size). IPTW: inverse probability treatment weighting; BMI: body mass index. * On a base-2 logarithmic scale.

**Table 2 cancers-15-00099-t002:** The linear and non-linear relationship between perioperative transfusion and recurrence-free and overall survival.

	Linear Effect			Nonlinear Effect		
	Estimate	SE	*p*	Estimate	SE	*p*
Recurrence-free survival	0.185	0.035	<0.001	−0.033	0.010	0.001
Overall survival	0.097	0.036	0.008	−0.005	0.009	0.557

**Table 3 cancers-15-00099-t003:** Stepwise model selection for recurrence-free survival before IPTW.

	HR	95% CI	*p*
Blood transfusion	1.40	0.99~1.97	0.059
Positive surgical margin	1.63	1.15~2.30	0.006
Primary tumor (T3,4 vs. T1,2)	1.59	1.12~2.27	0.010
Lymph node involvement	1.96	1.39~2.78	<0.001
Adjunct radiotherapy	0.61	0.42~0.88	0.007

HR: hazard ratio; CI: confidence interval.

**Table 4 cancers-15-00099-t004:** Forward model selection for overall survival before weighting.

	HR	95% CI	*p*
Blood transfusion	1.39	0.98~1.97	0.067
Age	1.02	0.84~0.94	0.002
BMI	0.94	1.12~1.37	0.002
Primary tumor (T3,4 vs. T1,2)	1.77	1.07~1.25	0.002
Lymph node involvement	3.44	1.85~5.24	<0.001

HR: hazard ratio; CI: confidence interval; BMI: body mass index.

## Data Availability

All data are available from the author directly.

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
