# Peer review of "Perioperative Blood Transfusion Is Dose-Dependently Associated with Cancer Recurrence and Mortality after Head and Neck Cancer Surgery"

_cancers, 2022, doi:10.3390/cancers15010099_

Round 1

Reviewer 1 Report

In their manuscript entitled "Perioperative Blood Transfusion is Dose-dependently Associated with Cancer Recurrence and Mortality after Head and Neck Cancer Surgery" the authors present a retrospective study regarding the effect of perioperative blood transfusion upon cancer recurrence and mortality. The manuscript is well written and will be of interest to the readership of the journal and the general scientific community. The fact that the authors performed the method of inverse probability of treatment weighting, really strengthens the outcome of this research and the authors should be praised for that.

This Reviewer only has some minor comments as follows:

1) I strongly believe that the review: 10.1016/j.transci.2017.05.015 should be mentioned in either the introduction or the discussion section of this manuscript since it thoroughly examines the cancer patients as blood recipients.

2) In the Discussion section, in the transfusion-related factors that have been proposed as impactful upon cancer patients, the authors should also mention the controversial role of extracellular vesicles that accumulate in the blood units. I hereby propose some relevant references, but the authors are free to support their additions however they want (10.3892/ijo.2021.5288 ; 10.3390/ijms23169323)

Author Response

Point 1:  I strongly believe that the review: 10.1016/j.transci.2017.05.015 should be mentioned in either the introduction or the discussion section of this manuscript since it thoroughly examines the cancer patients as blood recipients.

Response 1:

Thank you for providing the informative article "Red Blood Cell Transfusion in Surgical Cancer Patients: Targets, Risks, Mechanistic Understanding, and Further Therapeutic Opportunities." The review clearly illustrates the mechanisms at a cellular level and the clinical impact on various cancers. It will certainly provide our readers with a better understanding of this topic. The revised paragraph (paragraph 4 in the discussion section) is as follows:

The mechanism most commonly proposed as contributing to the influences of allogenic blood transfusion (ABT) on cancer outcome is the phenomenon known as transfusion-related immunomodulation(TRIM), which refers to the immune alterations associated with allogenic blood transfusion.[20] In a detail review article by Tzounakas et al, the authors described the underlying mechanisms of TRIM, including RBC storage lesions, the RBCs themselves, residual white blood cells, immunosuppressive cytokines, and biologic mediators.[21]

References

  1. Cata, J.P.; Wang, H.; Gottumukkala, V.; Reuben, J.; Sessler, D.I. Inflammatory response, immunosuppression, and cancer recurrence after perioperative blood transfusions. Br J Anaesth 2013, 110, 690-701, doi:10.1093/bja/aet068.
  2. Tzounakas, V.L.; Seghatchian, J.; Grouzi, E.; Kokoris, S.; Antonelou, M.H. Red blood cell transfusion in surgical cancer patients: Targets, risks, mechanistic understanding and further therapeutic opportunities. Transfus Apher Sci 2017, 56, 291-304, doi:10.1016/j.transci.2017.05.015.

Point 2: In the Discussion section, in the transfusion-related factors that have been proposed as impactful upon cancer patients, the authors should also mention the controversial role of extracellular vesicles that accumulate in the blood units. I hereby propose some relevant references, but the authors are free to support their additions however they want (10.3892/ijo.2021.5288 ; 10.3390/ijms23169323)

Response 2:

Thank you again for providing the informative articles. The revised paragraph is a shown below (paragraph 4 in the discussion section):

Another nontypical immunomodulator, the extracellular vesicles (EVs) is an RBC storage lesion which increases in amount as the RBCs age. EVs can be secreted by all cell types, and their biological functions include eliminating cel-lular waste, facilitating intercellular communication, and, of particular interest, modu-lating immune response.[22] The contents within EVs secreted by RBCs include RNAs, immunoglobulins, complement proteins, and exposed phosphatidylserine, are believed to actively play a role in activating TRIM in cancer patients.[23] The abovementioned components in the blood products could elicit immune responses that include both immunosuppressive and inflammatory effects.

References:

  1. Doyle, L.M.; Wang, M.Z. Overview of Extracellular Vesicles, Their Origin, Composition, Purpose, and Methods for Exosome Isolation and Analysis. Cells 2019, 8, doi:10.3390/cells8070727.
  2. Ma, X.; Liu, Y.; Han, Q.; Han, Y.; Wang, J.; Zhang, H. Transfusion‑related immunomodulation in patients with cancer: Focus on the impact of extracellular vesicles from stored red blood cells. International Journal of Oncology 2021, 59, 1-11.

Reviewer 2 Report

I've read with interest your paper titled "Perioperative Blood Transfusion..."

I really liked it and, as you say the main limitation is the nature of the study, a retrospective one, but I think there is another issue that should be improved. 

There is no mention about the type of red cell concentrate used in the study. As you mention in the discussion one of the possible mechanism related to the TRIM effect is the presence of leukocytes. That's why you should describe the type of red cell concentrates used, that is, buffy coat removed or not or leukorreduced or both. Furthermore it should have studied been studied (it is not mentioned) other blood component transfusion as they might have contributed to the TRIM effect by adding leukocytes (plateles for example)

Author Response

Point 1: I really liked it and, as you say the main limitation is the nature of the study, a retrospective one, but I think there is another issue that should be improved. 

There is no mention about the type of red cell concentrate used in the study. As you mention in the discussion one of the possible mechanism related to the TRIM effect is the presence of leukocytes. That's why you should describe the type of red cell concentrates used, that is, buffy coat removed or not or leukorreduced or both. Furthermore it should have studied been studied (it is not mentioned) other blood component transfusion as they might have contributed to the TRIM effect by adding leukocytes (plateles for example)

Response 1: Thank you for bringing this matter to our attention. The pRBC transfused in our study were nonleukodepleted pRBC. We did not further classify our pRBC based on the leukodepletion status of the blood products, which is indeed a limitation that should be addressed, along with the presence of other blood products. We have revised the manuscript to address this issue in the last paragraph. The revised paragraph is as follows:

Discussion-last paragraph

One limitation of our study is its retrospective nature, which may have contributed to nonrandomized and unstandardized interventions. While we carefully controlled for known confounding factors using inverse probability of treatment weighting (IPTW) analysis, residual confounding effects may still be present. Additionally, In this study, we did not examine the effect of leukodepletion on packed red blood cell (pRBC) transfusions. Most of the pRBC transfusions in our data were not leukodepleted. While the literature has examined the impact of leukocyte depletion on immunomodulation and cancer outcomes, both leukodepleted and non-leukodepleted pRBCs have been linked to poor cancer outcomes. Two randomized controlled trials found no association between leukocyte-depleted blood products and cancer recurrence or long-term sur-vival in colorectal cancer [40] and gastrointestinal cancer patients. [41] However, an-other study demonstrated that the transfusion of non-leukodepleted pRBCs was asso-ciated with ovarian cancer recurrence. [42] Moreover, we did not consider transfusion of other blood products such as fresh frozen plasma, platelet concentrates, platelet apheresis, etc. into account. Further study of leukodepleted status effects on cancer outcomes can be done in the future

References:

  1. van de Watering, L.M.; Brand, A.; Houbiers, J.G.; Klein Kranenbarg, W.M.; Hermans, J.; van de Velde, C. Perioperative blood transfusions, with or without allogeneic leucocytes, relate to survival, not to cancer recurrence. Br J Surg 2001, 88, 267-272, doi:10.1046/j.1365-2168.2001.01674.x.
  2. Lange, M.M.; van Hilten, J.A.; van de Watering, L.M.; Bijnen, B.A.; Roumen, R.M.; Putter, H.; Brand, A.; van de Velde, C.J. Leucocyte depletion of perioperative blood transfusion does not affect long-term survival and recurrence in patients with gastrointestinal cancer. Br J Surg 2009, 96, 734-740, doi:10.1002/bjs.6636.
  3. De Oliveira, G.S., Jr.; Schink, J.C.; Buoy, C.; Ahmad, S.; Fitzgerald, P.C.; McCarthy, R.J. The association between allogeneic perioperative blood transfusion on tumour recurrence and survival in patients with advanced ovarian cancer. Transfus Med 2012, 22, 97-103, doi:10.1111/j.1365-3148.2011.01122.x.
